# Study on Deflection-Span Ratio of Cable-Stayed Suspension Cooperative System with Single-Tower Space Cable

**Lin Xiao** , **Yaxi Huang** and **Xing Wei** *

School of Civil Engineering, Southwest Jiaotong University, Chengdu 610032, China;
xiaolin@home.swjtu.edu.cn (L.X.); yaxihuang1999@163.com (Y.H.)
* Correspondence: we_star@swjtu.edu.cn

**Abstract:** This study uses the wind–vehicle–bridge coupling vibration analysis method to investigate the bridge stiffness problem of a large-span cable-stayed-suspension cooperative system. On the basis of the particle-damping-spring vehicle model, the TMeasy surface contact tire model is introduced, and a set of universal wind–vehicle–bridge coupling analysis algorithm is built in the framework of the whole process iterative method. Based on the Latin supercube sampling principle, random traffic flow is generated and loaded onto bridge structures with different stiffness conditions to analyze the driving comfort and safety under each stiffness condition. Combining the specification requirements, engineering experience, and research results, the vertical stiffness limit applicable to the bridge of the highway cable-stayed-suspension collaborative system is proposed. Existing engineering experience shows that the vertical deflection-to-span ratio of a cable-stayed bridge under live load is distributed between 1/400 and 1/1600, and the vertical deflection span ratio under the action of lane load is recommended based on numerical analysis.

**Keywords:** cable-stayed-suspension cooperative system bridge; stiffness standards; wind-vehicle-bridge coupling vibration; tire model; driving comfort; driving safety

## 1. Introduction

The cable-stayed-suspension cooperative system bridge adds cable-stayed cables to the long-span suspension bridge to improve its wind resistance stability, which makes up for the lack of rigidity of the suspension system [1]. It also provides a new bridge form for mountainous and oceanic long-span bridges. At present, the long-span cable-stayed-suspension collaborative bridges in the world include the Third Bosphorus Bridge (main span 1408 m, two-way eight-lane high-speed + double-track railway). The Xihoumen Highway-Railway Bridge (main span 1488 m) and the Jingzhou Libu Yangtze River Highway-Railway Bridge (main span 1120 m) under construction in China also adopt the cable-stayed-suspension coordination system. The design scheme of a bridge across the Xunjiang River adopts a (638 + 638) m space cable single tower cable-stayed-suspension collaborative system scheme, the tower height is 231 m, and the spacing is 16 m, with a total of 19 m (Figure 1).

Both cable-stayed bridges and suspension bridges are classic cable-supported bridge systems, and the cable structure is prone to vibration. Bryja found that the cable car would exhibit unstable behavior under wind loads, such as violent swinging or sliding, when studying the interaction between aerial cable cars and cables [2]. The irregular shape of the contact line in the railway electrification system can lead to oscillation and resonance between the pantograph and the contact line, affecting the stable contact performance of the pantograph with the contact line [3]. Moreover, when the wind attack angle is at a specific value, the damping of the catenary becomes negative, and the running vibration reaches the maximum amplitude [4]. Both of the above structures are simple cable-supported structures, while bridge structures composed of multiple cables, such as cable-stayed and

suspension bridges, are prone to vibration behavior under dynamic loads [5]. Under wind loads, both cable-stayed and suspension bridges are prone to wind-induced vibration and flutter, but suspension bridges are more susceptible to nonlinear response due to their lower structural stiffness. The cable-stayed-suspension combined system bridge, as a combination of cable-stayed and suspension bridges, has a more complex response under wind and moving loads. Theoretical studies show that the structural nonlinear effect of cable-stayed-suspension cooperative system bridges is obvious, and the stiffness problem is more complicated than that of suspension bridges or cable-stayed bridges. The limit value of bridge stiffness involves the comprehensive consideration of structural safety and economy. At present, few studies examine the stiffness of bridges with cable-stayed-suspension cooperative system, and no relevant regulations are in place. Bridge stiffness mainly refers to its vertical stiffness and transverse stiffness [6], which are often measured by indicators such as vertical and transverse deflection–span ratios. China's "Code for Design of Highway Suspension Bridges" (JTGT D65-05-2015) stipulates that "the maximum vertical deflection value of stiffened beams caused by the frequent occurrence of lane loads should not be greater than 1/250 of the span." The "Code for Design of Highway Cable-Stayed Bridges" requires that the vertical deflection-span ratio shall not exceed 1/400. Japan's "Road and Bridge Indication Book" specifies that the deflection–span ratio of cable bridges shall not exceed 1/350. Bridge stiffness affects the dynamic response of vehicles passing on the bridge. Reasonable bridge stiffness should take the comfort and safety of driving on the bridge into consideration [7]. Given the space of a cable-stayed-suspension cooperative system bridge in Figure 1, this study uses the wind–vehicle–bridge coupling analysis as a means to establish a tire mechanics model considering the tire deformation surface. This study also explores the vertical deflection–span ratio limit under the effect of satisfying driving comfort. The value provides a reference for the limit value of the bridge stiffness of the cable-stayed-suspension cooperative system. As the transverse stiffness of the space cable structure is greater than that of the parallel cable structure, the study mainly discusses the vertical stiffness of the bridge.

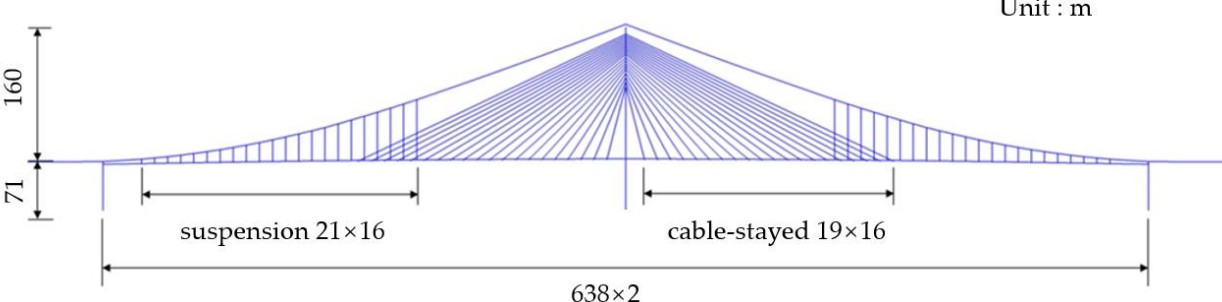

**Figure 1.** Schematic diagram of cable-stayed-suspension cooperative system with single tower of space cable.

## 2. Evaluation Method of Bridge Driving Comfort and Safety

Driving comfort and safety are directly related to vehicle dynamic response. Based on the coupled vibration analysis theory of wind–vehicle–bridge, the dynamic interaction between vehicle and bridge can be considered, and the response of bridge structure and vehicle can be accurately solved. It is an effective method for bridge stiffness analysis and evaluation, which is widely used in railway bridge engineering [8]. In particular, it is applied in the setting of safe wind speed and safe driving speed [9], the evaluation and optimization of wind barrier performance [10], the research on bridge structure vibration and vibration reduction measures [11], and the evaluation of track irregularity of high-speed railways [12]. However, the application research in highway bridges is relatively scarce. Basing on the design load of Chinese bridge code, investigation and sampling results, and equivalent theory, Deng Luji put forward a vehicle model suitable for wind–vehicle–bridge coupling calculation in China, including the values of geometric dimensions, mass,

stiffness, and damping, and formed a systematic vehicle model library [13]. Polish scholars established high-precision vehicle models through the multi-body dynamics simulation software ADAMS and LS-DYNA. This kind of model has a strong application significance for the response of the car body, but its calculation efficiency is too slow to be suitable for the large-scale windmill-bridge coupling analysis [14].

*2.1. Dynamic Model of Bridges and Vehicles*

The bridge model is simplified as a beam model, and the bridge model is established in the finite element analysis software ANSYS (Figure 2). The beam element Beam4 is used for the main girder and tower, whereas Link10 is used for the simulation of the main cable, suspenders, and cables. The coupling function of degrees of freedom is used at the connection of tower and beam to constrain the longitudinal torsion, longitudinal translation, and lateral translation of the main beam. The end of the main girder constrains the longitudinal, transverse, and torsional degrees of freedom around the longitudinal axis. Both the rod element and the beam element are shown in blue, and the constraints are shown in green. In the vehicle–bridge coupling analysis, the vehicle is usually abstracted as a particle-damping-spring model, in which rigid bodies, such as car body, axle, and wheel, are connected with one another through dampers and elastic elements. The motion equation of the vehicle is constructed using the D'Alembert principle. The vehicle suspension system and wheel are abstracted as spring-damping elements, and the vehicle mass is assumed to be distributed in the center of mass of each rigid body. The car body has six degrees of freedom in space, namely, floating, swinging, stretching, nodding, shaking, and rolling. However, because the longitudinal vibration of the car along the driving direction has little influence on the vertical and lateral vibration of the bridge, the longitudinal freedom of the car body, that is, the degree of freedom of stretching, is generally ignored [15].

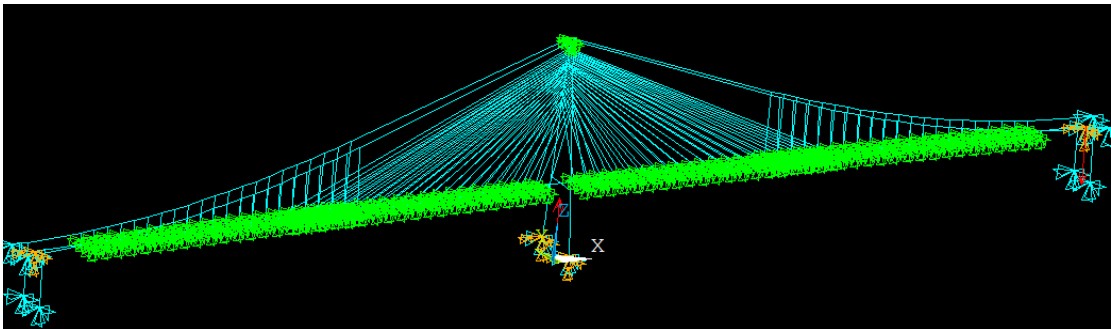

**Figure 2.** Finite element model diagram of bridge.

The difficulty in the analysis of vehicle–bridge coupling vibration is the contact model between the wheel and the bridge deck, and the key is the tire model. Wheel motion parameters include longitudinal slip rate $s$, slip angle $\alpha$, radial deformation $\rho$, camber angle $\gamma$, wheel speed $\omega$, and yaw angle $\beta_t$. According to different research purposes, tire dynamics establishes tire longitudinal sliding model, tire cornering model, and tire vertical vibration model. According to the applicable analysis state, it is subdivided into steady-state model, dynamic model, and tire model for the dynamic simulation [16]. TMeasy tire model is a tire model provided by Simpack software, which is suitable for dynamic simulation. The calculation of the sideslip force of the tire comes from the Lugre dynamic model [17], and the vertical stiffness nonlinear model is used. The vertical load is divided into two parts: static load $F_z^{st}$ and dynamic load $F_z^{D}$. The static load is expressed as a nonlinear function related to vertical displacement, and the dynamic load is approximately expressed as a linear damping model, namely:

$$F_z = F_z^{st} + F_z^{D} = a_1 \Delta z + a_2 (\Delta z)^2 + d_T \Delta \dot{z} \tag{1}$$

where $F_z$ is the vertical load of the tire; $\Delta z$ is the vertical displacement of the tire; $\Delta \dot{z}$ is the derivative of vertical displacement; $a_1$ and $a_2$ are the effective load and the radial stiffness at twice the effective load, respectively; and $d_T$ is the vertical damping of the tire.

$$F_y = c_y \Delta y + d_y \Delta \dot{y} \qquad (2)$$

where $F_y$ is the lateral force of the wheel, $c_y$ is the lateral stiffness of the tire, $d_y$ is the lateral damping of tires, and $\Delta y$ is the lateral deformation of the tire.

Taking a two-axle vehicle as an example, the vehicle space dynamics model is established in Simpack, and its topology model and entity model diagram are shown in Figures 3 and 4, respectively. In Figure 3, the blue part represents the degree-of-freedom coupling, the red part represents the force element, and the different numbers represent the different force elements; number 5 in the figure represents the spring-damping force element, number 253 represents the TMeasy force element, and the green part is the constraint equation.

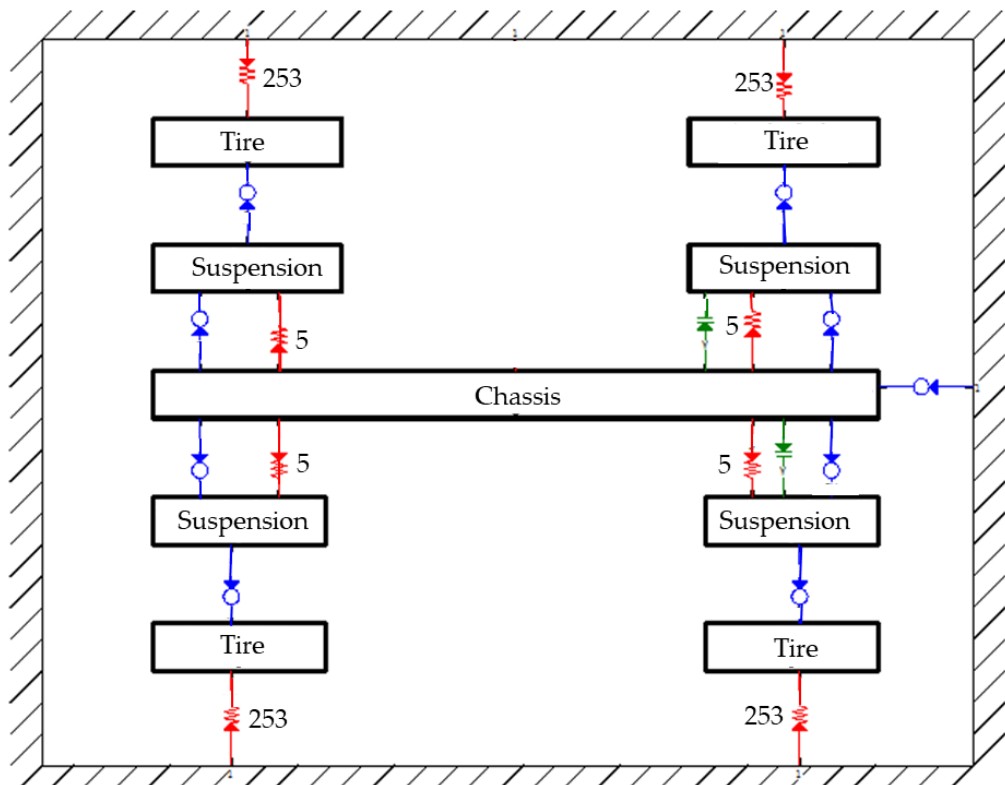

**Figure 3.** Topological model diagram of two-axle vehicle.

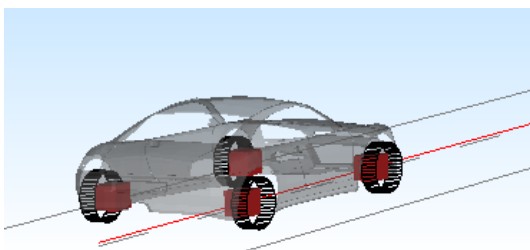

**Figure 4.** Solid model diagram of two-axle vehicle.

### 2.2. Effect of Fluctuating Wind Field on Vehicle–Bridge System

The wind-induced response of the bridge considers the resistance, lift, torque, and buffeting force of the stiffening beam caused by the average wind. The average wind-induced

response is taken according to the Code for Wind-resistant Design of Highway Bridges, and the buffeting force is calculated by considering the unsteady and local spatial correlation of buffeting force on the basis of the Scanlan quasi-steady aerodynamic formula [18]. The existing aerodynamic admittance function is usually a function in frequency domain, which cannot be directly used to solve the buffeting force of bridges in time domain. Usually, the equivalent wind spectrum method and buffeting force spectrum method can be used for time domain analysis. The cross-section of bridge structure is very complex and often blunt, so obtaining the accurate expression of aerodynamic admittance function of specific cross-section is difficult at this stage [19]. In the absence of accurate aerodynamic admittance function, the simplified flat aerodynamic admittance function expression proposed by Liepmann is generally used for flat streamlined sections, whereas the aerodynamic admittance function can be safely set to one for relatively passive bridge structural sections without considering the influence of aerodynamic admittance. The aerodynamic effect of wind on bridge is shown in Figure 5 and the buffeting resistance, buffeting lift, and buffeting moment per unit length of main girder are expressed as follows:

$$F_{b,d}^H = \rho UB \left\{ \frac{H}{BC_{H,d}(\alpha)\chi_{Hu}u(t)} + 1/2[H/B \frac{dC_{H,d(\alpha)}}{d\alpha} \chi_{Hw}w(t)] \right\} \tag{3}$$

$$F_{b,d}^v = \rho UB \left\{ C_{V,d}(\alpha)\chi_{Vu}u(t) + +1/2[\frac{dC_{H,d(\alpha)}}{d\alpha} + H/BC_{H,d}(\alpha)]\chi_{Vw}w(t) \right\} \tag{4}$$

$$F_{b,d}^M = \rho UB^2 \left[ C_{M,d}(\alpha)\chi_{Vu}u(t) + +1/2 \frac{dC_{M,d(\alpha)}}{d\alpha} \chi_{Mw}w(t) \right] \tag{5}$$

where $F_{b,d}^H$, $F_{b,d}^v$, and $F_{b,d}^M$ are buffeting resistance, buffeting lift, and buffeting moment of main girder caused by pulsating wind, respectively; $u$ and $w$ are horizontal and vertical fluctuating wind speed components; $d/d\alpha$ is the derivation of wind attack angle; $\chi_{Hu}$, $\chi_{Hw}$ are aerodynamic admittance functions related to buffeting resistance; $\chi_{Vu}, \chi_{Vw}$ are aerodynamic admittance functions related to buffeting lift; and $\chi_{Mu}$, $\chi_{Mw}$ are aerodynamic admittance functions related to buffeting torque.

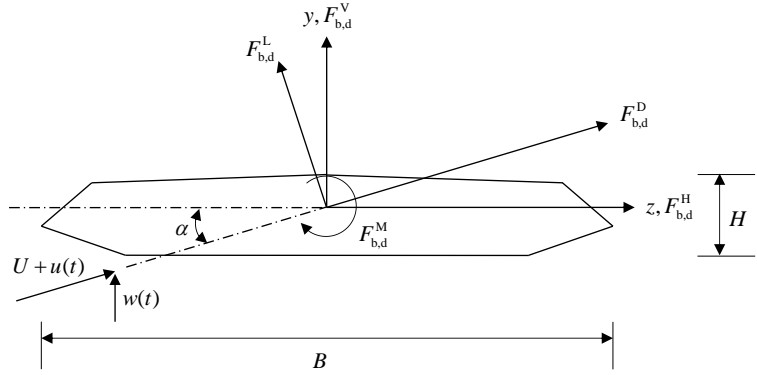

**Figure 5.** Aerodynamic effect of wind on bridge.

The wind load borne by vehicles is usually described by Baker's expression based on quasi-steady theory:

$$F_S = 0.5\rho U_r^2(t)C_S(\Psi)A_0 \tag{6}$$

$$F_L = 0.5\rho U_r^2(t)A_0 C_L(\Psi) \tag{7}$$

$$M_R = 0.5\rho U_r^2(t)C_R(\Psi)A_0 h_v \tag{8}$$

where $F_S$, $F_L$, and $M_R$ are the lateral force, lifting force, and overturning moment acting on the center of mass of the car body, respectively; $C_S(\psi)$, $C_L(\psi)$, and $C_R(\psi)$ are the lateral force, lift, and overturning moment coefficients of the vehicle, respectively, which are obtained by CFD test; $A_0$ is the windward area of the vehicle; $h_v$ is the distance from the center of mass

of the car body to the road surface; $U_r$ is the relative speed between the vehicle and the wind; and $\psi$ is the corresponding deflection angle. Assume that the wind with the speed of $U$ acts vertically on the longitudinal axis of the road, and the vehicle travels at the speed of $U_v$. The relative speed and its deflection angle are calculated according to the following formula, where $u(x,t)$ represents the turbulent wind acting on the vehicle at time $t$.

$$U_r = \sqrt{(U + u(x,t))^2 + U_y^2} \tag{9}$$

$$\psi = arctan\big((U + u(x,t))/U_y\big) \tag{10}$$

*2.3. Vehicle–Bridge Interaction and Solution*

According to the static and dynamic interaction and displacement coordination among vehicle subsystem, bridge subsystem and wind, the vibration equation of a vehicle–bridge–wind coupling system can be established as shown in Equations (11) and (12):

$$M_c \ddot{X}_c + C_c \dot{X}_c + K_c X_c = F_c^b + F_c^w \tag{11}$$

$$M_b \ddot{X}_b + C_b \dot{X}_b + K_b X_b = F_b^c + F_b^w \tag{12}$$

where $M_c$, $C_c$, and $K_c$ are the mass, damping, and stiffness matrices of the vehicle, respectively; $F_c^b$ is the force of the bridge on the vehicle; $F_c^w$ is the force of the wind on the vehicle; $M_b$, $C_b$, and $K_b$ are the mass, damping, and stiffness matrices of the bridge, respectively; $F_b^c$ is the force of the vehicle on the bridge; $F_b^w$ is the force of the wind on the bridge; $X_c$ is the displacement of the vehicle center of mass; and $X_b$ is the displacement at the contact point of the vehicle axle.

The irregularity of the bridge deck is regarded as a spatial random function $z = z(x,y)$, and the position and direction of the tire are respectively represented by the position vector $r_{0C}$ and the unit vector $e_{yR}$. The geodetic coordinate system is taken as the reference system (Figure 6b), where $r_{0C}$ is the vector formed by connecting the point C of the tire centroid with the origin of the geodetic coordinate system, indicating the displacement of the tire centroid relative to the geodetic coordinate system. The unit vector $e_{yR}$ determines the direction of the tire plane.

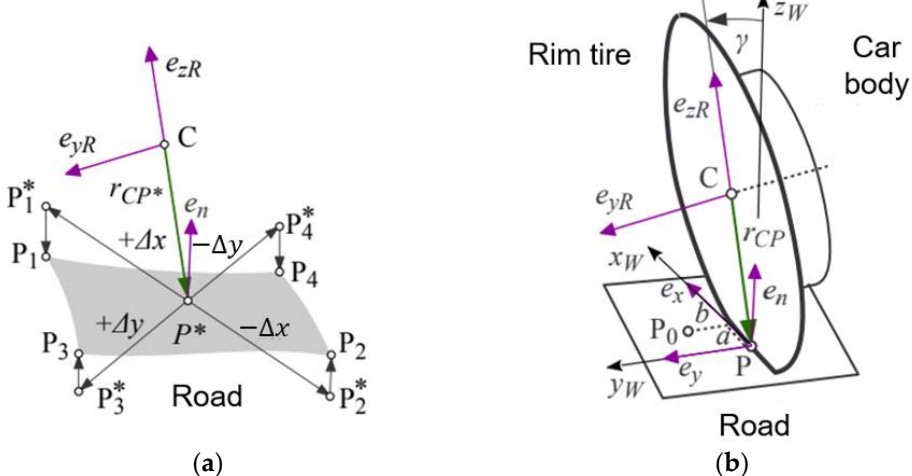

**Figure 6.** (**a**) Spatial relation diagram of P point; (**b**) tire coordinate system.

When the tire is on an uneven road surface, the position of the projection point P of the tire center of mass on the ground along the tire center plane cannot be directly calculated. To approximate the position of the contact point, the wheel–ground contact surface is assumed to be a plane, which is determined by the road conditions where the tire is located. Figure 6a shows a schematic diagram of this calculation method, taking the tire

centroid *C* as the center point and the plane passing through point *C* and perpendicular to the tire center plane as the reference plane, a rectangle with a semi-longitudinal length of $\Delta x = 0.4 \times r_0$ and a semi-transverse length of $\Delta y = 0.4 \times b_0$ is selected, where $r_0$ and $b_0$ represent the unloaded state. The intersection of the four vertices of the rectangle with the road surface along the tangent direction line of the tire center plane is the road surface characteristic point. If the four feature points are not in the same plane, the elevation of the feature points is automatically adjusted to place them in the same plane. The normal vector $e_n$ of the wheel pavement plane determines the Z-axis of the track at this moment, and the intersection of the tire center plane and the pavement determines the X-axis and Y-axis of the track at this moment. The midpoint of this intersection is the P-point, hereinafter referred to as the geometric contact point (Figure 7).

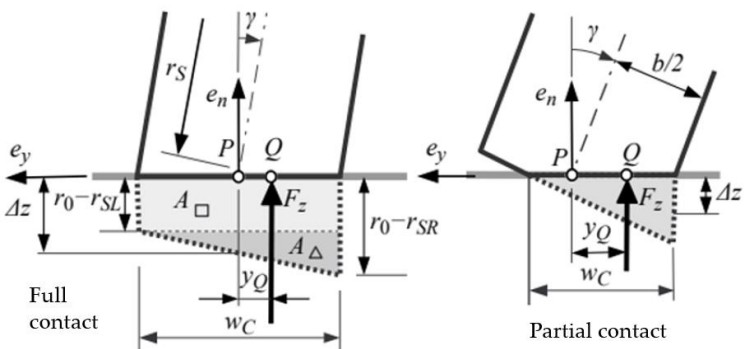

**Figure 7.** Schematic diagram of wheel–ground contact surface.

When solving, the bridge subsystem is assumed to be rigid, and the vehicle motion and vehicle force time history are obtained by solving the independent vehicle equation. Then, the wheel force is applied to the bridge, and the bridge deck motion state is obtained by solving the independent bridge equation. The superposition of the bridge deck motion time history and bridge deck irregularity is used as a new vehicle system excitation for the next iteration until the force between the vehicle and the bridge deck converges (Figure 8).

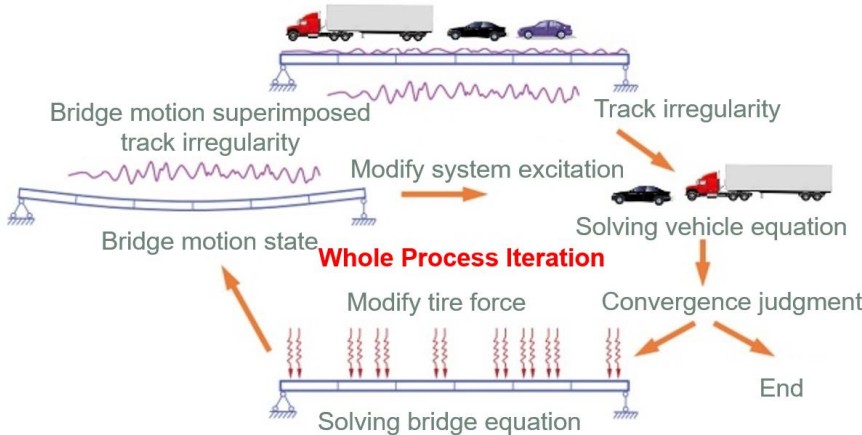

**Figure 8.** Schematic diagram of whole process iteration method.

### 2.4. Driving Comfort and Safety

Comfort and safety evaluation is based on the vehicle–bridge coupling vibration theory, which is an evaluation index that meets the human sensory and safety requirements. The most widely used comfort evaluation method is the 1/3 frequency doubling method [20] and the total weighted acceleration method (RMSV) proposed in ISO2631 standard. In recent years, the RMSV method has been widely used in research [21], which is more comprehensive than the frequency doubling method. The former obtains weighted

acceleration root mean square value $a_w$ by considering multi-directional vibration after weighting filtering the vehicle acceleration time domain signal. Its calculation formula is as follows:

$$a_w = \left[ \int_{0.9}^{80} W^2(f) G_a(f) df \right]^{0.5} \tag{13}$$

where $W(f)$ is the frequency weighting function, and there are different weighting functions in ISO2631 standard for different vibration directions; $G_a(f)$ is the power spectral density function of the acceleration time domain signal.

Judging human comfort by $a_w$ value (Table 1), this study uses 0.315 m/s² as the comfort limit for subsequent analysis.

**Table 1.** Weighted acceleration grading table.

| Level | $a_w$ (m/s²) | Comments |
|---|---|---|
| 1 | <0.315 | Comfortable |
| 2 | 0.315–0.630 | A little uncomfortable |
| 3 | 0.500–1.000 | Less comfortable |
| 4 | 0.800–1.600 | Uncomfortable |
| 5 | 1.250–2.500 | Very uncomfortable |
| 6 | >2.000 | Extremely uncomfortable |

The roll safety factor RSF and sideslip safety factor SSF are often used as evaluation indicators [22]. The evaluation criteria of vehicle roll accident in the Specification for Dynamic Performance Evaluation and Test Appraisal of Locomotive and Rolling Stock (GB/T 5599-2019) are as follows:

$$\text{RSF} = \min\left\{ \left| \frac{\sum_{i=1}^{k} (F_{Li} + F_{Ri})}{\sum_{i=1}^{k} (F_{Li} - F_{Ri})} \right| \right\} \geq 1.2 \tag{14}$$

where $F_{Li}$ is the axle load of each wheel on the left, and $F_{Ri}$ is the axle load of each wheel on the right.

The sideslip resistance of a vehicle is defined as:

$$F_{SR} = \mu_s (F_{vl} + F_{vr}) - (F_{hl} + F_{hr}) \tag{15}$$

where $F_{vl}$ and $F_{vr}$, respectively, represent the vertical contact forces of the windward wheel and the leeward wheel of an axle, and $F_{hl}$ and $F_{hr}$, respectively, represent the lateral contact forces of the windward wheel and the leeward wheel of an axle; $\mu_s$ is the lateral attach rate between the wheel and the road surface, which can be set to 0.7, 0.5, 0.15, and 0.07 according to the road conditions, representing the four basic road conditions of dry, wet, snow, and ice, respectively.

When the vehicle has the tendency of lateral sliding, the vehicle is considered to have already a sideslip safety accident. Therefore, the dimensionless SSF after considering the safety reserve factor can be defined as:

$$\text{SSF} = \frac{\overline{F_{SR}} - 1.645\sigma_{SR}}{0.2\mu_s G_a} \geq 1.0 \tag{16}$$

where $G_a$ is the gravity of an axle of a vehicle, and generally the axle with light axle weight is taken; 0.2 is the safety reserve factor; $\overline{F_{SR}}$ is the average value of vehicle side-slip resistance, and $\sigma_{SR}$ is the variance of vehicle side-slip resistance.

## 3. Simulation of Random Traffic Flow Based on Latin Hypercube Sampling

Sampling is a universal means to obtain random samples. For random variables with specific distribution, the Monte Carlo (MC) sampling method or Latin hypercube sampling method (LHS) can be used [23]. It is difficult for the random variables generated by the

MC method to approximate the actual probability distribution accurately when the sample size is small. In contrast, LHS sampling can effectively reduce the complexity of calculation by layering the probability and then sampling layer by layer, and its approximation effect is obviously better than that of the MC method when the sample size is small [24].

Taking one-dimensional LHS sampling as an example (Figure 9), the cumulative distribution curve is divided into several equal intervals through stratification. Only one sample is randomly selected in each stratification to form a group of one-dimensional random variables, which is suitable for single parameter sample sampling. LHS sampling is a memory-related sampling method, which takes into account the samples that have been used before, and it has a good approximation effect. It is suitable for the simulation of random traffic flow with a sample size of 10 digits and 100 digits.

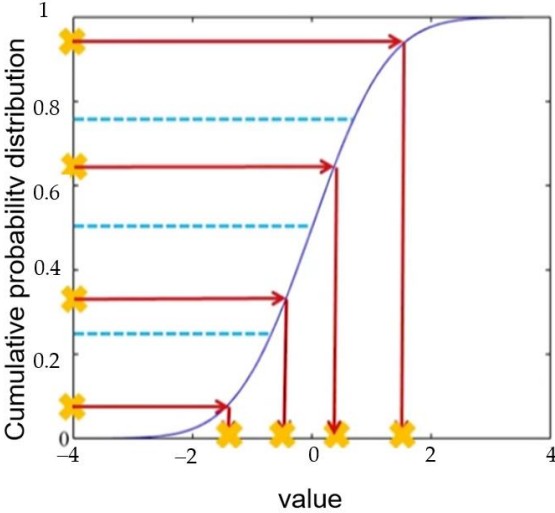

**Figure 9.** Schematic diagram of Vera Ding Chao cubic sampling.

Random traffic flow is mainly composed of four parts: the number of vehicles, the distribution of vehicle types, the distribution of axle load and wheelbase, and the distribution of vehicle speed and vehicle weight. Traffic volume forecast usually converts motor vehicles into the equivalent traffic volume of a standard vehicle according to their occupation of the road when driving on the road for statistical study. Generally, it is converted into the number of standard cars (pcu). According to the road traffic forecast report, the fleet with a total number of vehicles of 30 pcu is selected as the total number of sampling samples. LHS is used to stratify the distribution of vehicle type, axle load wheelbase, speed, and vehicle weight. Among them, the proportion of vehicle types conforms to the uniform distribution (Table 2), the distribution of axle load and wheelbase conforms to the lognormal distribution, and the distribution of vehicle speed and distance conforms to the normal distribution [25]. The axle load distribution of V1 car generated by Python language programming is shown in Figure 10, and the typical random fleet pattern in six lanes is shown in Figure 11.

**Table 2.** Vehicle distribution table.

| Symbol | Category | Ratio | Conversion Coefficient |
|--------|----------|-------|------------------------|
| V1 | Car | 40.6% | 1 |
| V2 | Bus | 5.8% | 2.0 |
| V3 | Buggy | 13.8% | 2.0 |
| V4 | Medium truck | 10.9% | 3.0 |
| V5 | Big truck | 10.1% | 4.0 |
| V6 | Trailers | 10.2% | 4.0 |
| V7 | Container truck | 8.6% | 4.0 |

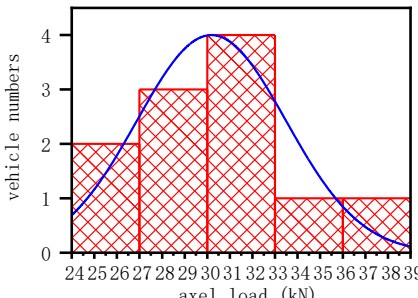

**Figure 10.** Axle load distribution diagram of V1 vehicle.

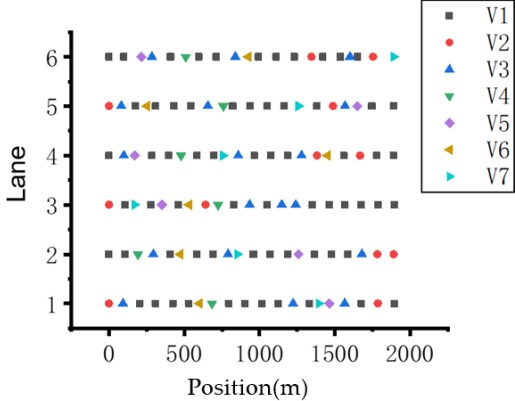

**Figure 11.** Random traffic flow simulation diagram.

## 4. Bridge Driving Safety and Comfort Analysis for Cable-Stayed-Suspension Cooperative System

### 4.1. Conditions

Aiming at the cable-stayed suspension bridge in Figure 1, an analysis model is established based on the aforementioned wind–vehicle–bridge coupling analysis method to study the influence of bridge stiffness change on driving safety and comfort.

The bridge is a single tower cable-stayed suspension cooperative system bridge with a main span of 2 × 638 m, and the main cable span is 2 × 730 m. Each side of the pylon is provided with 20 pairs of stay cables and 21 pairs of ordinary slings, in which the stay cables are arranged in a harp shape. The longitudinal anchorage spacing between the slings and stay cable beams is 16 m, and the crossing section spacing is 8 m, which is staggered. The length of the main girder of the single-span stay cable section is 327 m, and the length of the main girder of the sling section is 351 m. The cross section of the main girder is 26 m wide and 3 m high, and its standard section layout is shown in Figure 12.

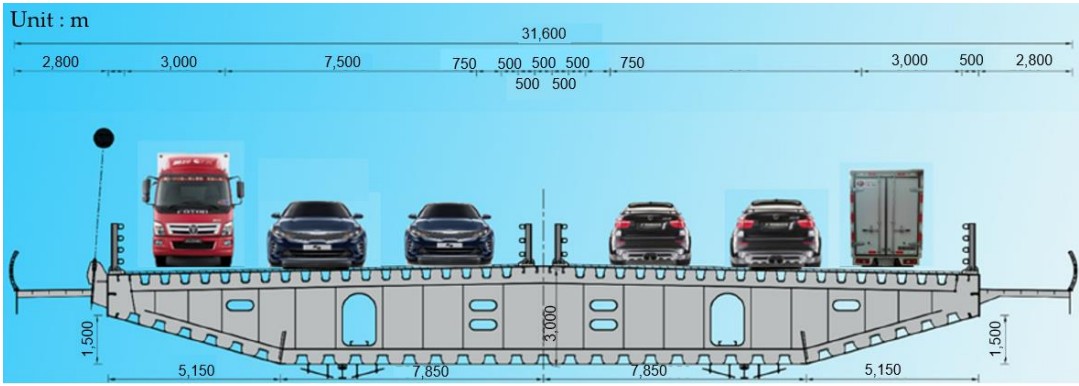

**Figure 12.** Layout of a standard bridge section.

To facilitate parameter analysis, based on the vertical static deflection span ratio of 1/330 under the load of two-way six lanes in the preliminary design scheme, the vertical deflection span ratio is adjusted by reducing the elastic modulus of the main girder, main cable, and stay cable, as shown in Table 3. Considering the driving situation in actual operation, two kinds of driving conditions, namely, ordinary driving conditions and extreme driving conditions, are compared under each torsion–span ratio. Ten kinds of analysis conditions are obtained. Among them, under normal driving conditions, random traffic loads are arranged on two-way six lanes, with 20 vehicles (30 pcu) in each lane, of which the proportion of heavy vehicles is 60%, and the maximum speed is 120 km/h. Under extreme driving conditions, lane loads (static loads) are arranged in five lanes as simulated traffic jams according to design specifications. Random traffic flow (20 vehicles, 120 km/h) is arranged in the sixth lane for loading to obtain dynamic response. In the analysis, the bridge deck wind speed is considered as the design wind speed of 25 m/s, and the pavement grade is Grade B.

**Table 3.** Stiffness parameters of bridge under extreme working conditions.

| Conditions Number | Deflection–Span Ratio under Lane Load |
| --- | --- |
| 1 | 1/330 |
| 2 | 1/300 |
| 3 | 1/280 |
| 4 | 1/250 |
| 5 | 1/200 |

Figure 13 shows the time history of vertical displacement in the left span under ordinary driving conditions (six-lane random traffic flow). Table 4 shows the static deflection and dynamic deflection under the same load. The influence of vehicle–bridge coupling vibration on deflection is analyzed by examining the vertical static displacement in the left span corresponding to vehicle load when the vertical displacement is maximum. The figure shows that the vertical dynamic deflection of the bridge increases gradually with the decrease of stiffness. When the bridge stiffness drops to 1/200, the vertical response value increases obviously because the whole bridge softens.

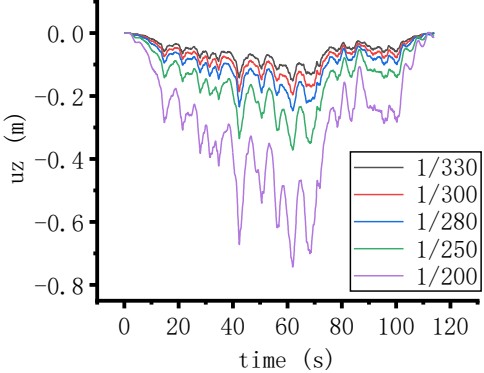

**Figure 13.** Time history of vertical displacement in the middle span of left girder under ordinary driving conditions.

Figure 14 shows the vertical dynamic displacement time history of the left span under extreme driving conditions (i.e., the static deflection caused by the load of five lanes is deducted). Table 5 shows the static deflection and dynamic deflection under the same load. From the head car to the bridge tower, the vertical displacement in the middle of the span first increases and then decreases to zero level. As the team drives to the right half span, the middle node of the left span is warped by the right load and presents a periodic vibration trend. The variation trend of the vertical response of the bridge under various stiffness conditions is roughly the same as that under ordinary driving conditions.

**Table 4.** Comparison table of static and dynamic displacement caused by vehicles under ordinary driving conditions.

| Conditions Number | Maximum Displacement in Left Span (cm) | |
| --- | --- | --- |
| | Vehicle-Induced Static Displacement | Coupled Vibration Displacement |
| 1 | 13.79 | 15.18 |
| 2 | 17.94 | 19.63 |
| 3 | 22.87 | 24.67 |
| 4 | 34.54 | 37.14 |
| 5 | 67.28 | 74.27 |

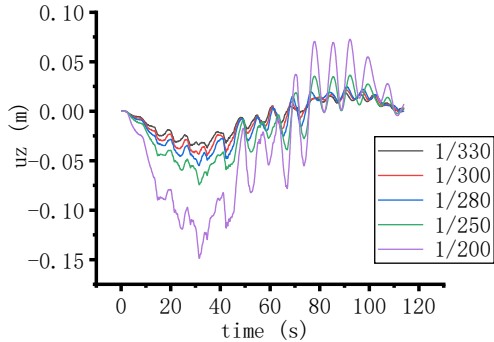

**Figure 14.** Time-history diagram of vertical displacement of left span.

**Table 5.** Comparison table of vehicle-induced static and dynamic displacement under extreme driving conditions.

| Condition Number | Maximum Displacement in Left Span (cm) | |
| --- | --- | --- |
| | Vehicle-Induced Static Displacement | Coupled Vibration Displacement |
| 1 | 2.7 | 3.71 |
| 2 | 4.37 | 4.46 |
| 3 | 5.32 | 5.50 |
| 4 | 6.35 | 7.44 |
| 5 | 12.35 | 14.88 |

*4.2. Vehicle Dynamic Response Analysis*

Under normal driving conditions, the vertical acceleration time history of the center of mass of the vehicle in the most unfavorable driving state (the driving speed is 120 km/h) is shown in Figure 15. When the vehicle is driving in the pure suspension section, the acceleration change is not obvious. When the vehicle approaches the transition section of cable-stayed suspension, the ride comfort of the bridge deck worsens, and the vertical acceleration of the vehicle increases owing to the change of structural stiffness. When driving to the cable-stayed section, the acceleration changes tend to be gentle, the displacement at the bridge tower is very small due to the constraint, and the corner at the junction of the tower and beam is discontinuous under the action of vehicle load. Therefore, the vertical acceleration of the vehicle here increases obviously. The vertical acceleration trend of vehicles under each working condition is basically the same. Owing to the large response of the bridge under 1/200 working condition, the corresponding vehicle acceleration response is larger than that of other vehicles.

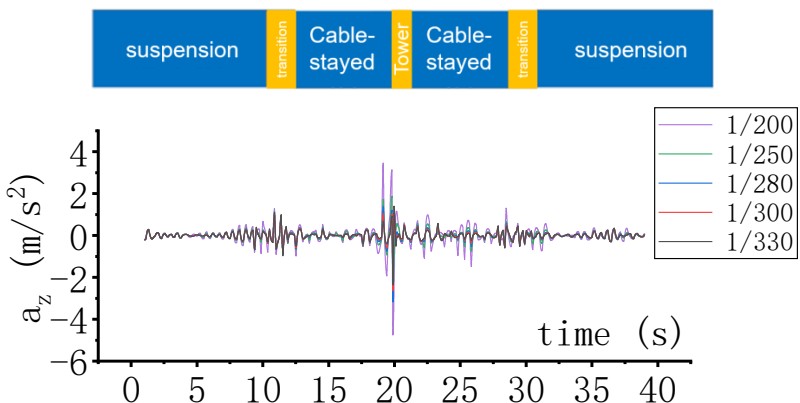

**Figure 15.** Vertical acceleration time history of vehicles under normal driving conditions.

The vertical acceleration response of vehicles in the most unfavorable situation in the fleet under extreme working conditions is shown in Figure 16. Owing to the large deformation of the bridge beam under static and live load, the driving alignment of vehicles under each working condition is quite different, and its curvature changes faster than that under ordinary driving conditions. As a result, the acceleration response of vehicles under extreme working conditions is obviously greater than that under ordinary driving conditions. When the bridge stiffness decreases, because the suspension cable segment itself is more flexible, its ability to resist deformation is weaker than that of the cable-stayed segment when the overall stiffness decreases uniformly. Therefore, the greater the overall stiffness decreases, the greater the deflection ratio corresponding to vehicle load, the more obvious the deformation of the suspension cable segment, and the greater the vertical acceleration response of the corresponding vehicle. When the vehicle travels to the cable-stayed section, the vertical acceleration response also increases with the decrease of stiffness, but its increase is weaker than that of the cable-stayed section.

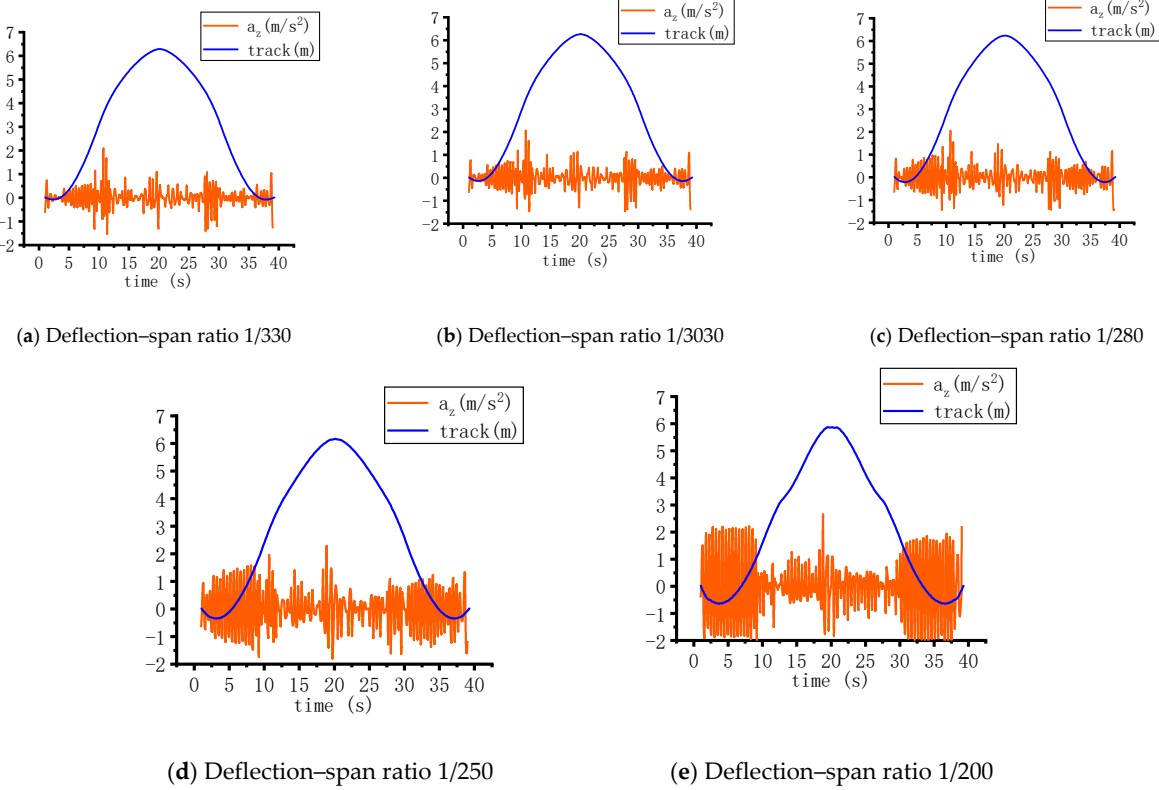

**Figure 16.** Vertical acceleration time history of vehicles under extreme driving conditions.

Compared with ordinary driving conditions, the peak value of vertical acceleration of the vehicle is smaller under extreme working conditions. At this time, because there is only one lane under extreme working conditions, the load value is small and the corresponding dynamic displacement is smaller. The main reason the dynamic response of vehicles under extreme driving conditions is generally greater than that under ordinary driving conditions in other sections is that the initial alignment of the bridge is deteriorated owing to the full load on the driveway under extreme driving conditions. In addition, the main girder alignment in the sling area and stay cable area is quite different, which increases the deck irregularity. As the self-weight of random traffic increases the irregularity of the bridge deck, to reduce the interference of self-weight, the single-lane load in extreme working conditions is replaced by a V1 vehicle with a self-weight of 2.4 t, so as to explore the influence of bridge alignment change on the vertical weighted acceleration of the vehicle. The vertical acceleration time history of vehicles under various working conditions is obtained by analysis. The weighted acceleration root mean square value is compared with the extreme working conditions, as shown in Table 6:

**Table 6.** Root mean square value of vertical acceleration under extreme driving conditions $a_z$ (m/s$^2$).

| Deflection–Span Ratio | Single Car | Random Traffic Flow |
| :---: | :---: | :---: |
| 1/330 | 0.108 | 0.242 |
| 1/300 | 0.110 | 0.239 |
| 1/280 | 0.125 | 0.263 |
| 1/250 | 0.180 | 0.320 |
| 1/200 | 0.516 | 0.514 |

The change of static alignment has an obvious influence on the root mean square value of vertical acceleration of vehicles. When the deflection is less than 1/250, the bridge alignment is relatively smooth, and the comfort decreases slightly with the increase of deflection. When the stiffness decreases to 1/200 corresponding to vehicle load (10.5 kN/m per lane), the linear deformation of the bridge is serious, and the driving comfort is seriously reduced.

### 4.3. Evaluation of Driving Safety and Comfort and Limit Value of Bridge Deflection Ratio

According to Formulas (13) to (16), the comfort and safety indexes of the vehicle are obtained after the vehicle response processing, as shown in Tables 7 and 8. Under normal driving conditions, the driving safety of vehicles meets the requirements, and the comfort is more sensitive to the change of vertical stiffness than safety. When the vertical deflection ratio is controlled below 1/250, the driving comfort still meets the requirements of the specification. When the vertical deflection ratio reaches 1/200, the vehicle still has a high safety reserve, but its driving comfort exceeds the limit.

**Table 7.** Comfort and safety index of vehicles under ordinary driving conditions.

| Torsion Span Ratio | $a_w$ (m/s$^2$) | RSF | SSF |
| :---: | :---: | :---: | :---: |
| 1/330 | 0.188 | 2.688 | 2.666 |
| 1/300 | 0.192 | 2.681 | 2.663 |
| 1/280 | 0.236 | 2.677 | 2.652 |
| 1/250 | 0.291 | 2.584 | 2.637 |
| 1/200 | 0.406 | 2.336 | 2.544 |

Compared with ordinary driving conditions, the driving safety and comfort of vehicles will decline under extreme driving conditions, but the safety index is still in a reasonable range. When the vertical torsion span ratio is greater than 1/280, the root mean square value $a_w$ of vehicle weighted acceleration reaches 0.315, which no longer meets the comfort requirements.

**Table 8.** Vehicle comfort and safety index under extreme driving conditions.

| Torsion Span Ratio | $a_w$ (m/s$^2$) | RSF | SSF |
| --- | --- | --- | --- |
| 1/330 | 0.278 | 2.295 | 2.477 |
| 1/300 | 0.293 | 2.245 | 2.455 |
| 1/280 | 0.315 | 2.231 | 2.446 |
| 1/250 | 0.41 | 2.184 | 2.434 |
| 1/200 | 0.639 | 2.158 | 2.247 |

As can be seen from the above, for the cable-stayed cooperative system bridge studied here, the vehicle driving safety index is easier to meet under each deflection–span ratio, and the driving comfort becomes the limiting factor of the deflection–span ratio. The comparison of comfort indexes under various working conditions is shown in Figure 17. From the point of view of driving comfort, the bridge deflection ratio must be controlled to not be greater than 1/280.

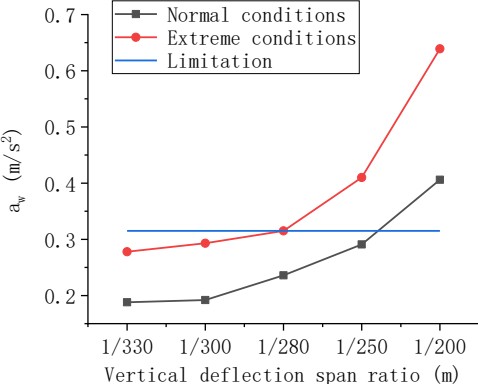

**Figure 17.** Correspondence between ride comfort and vertical deflection–span ratio.

The distribution of vertical stiffness index of the bridge under construction and completed cable-stayed cooperative system is shown in Figure 18, the blue signs represent the values. The figure shows that the vertical deflection span ratio of the cable-stayed cooperative system in the existing projects is between 1/619 and 1/402 [1,26,27], mostly for highway-railway dual-purpose bridges.

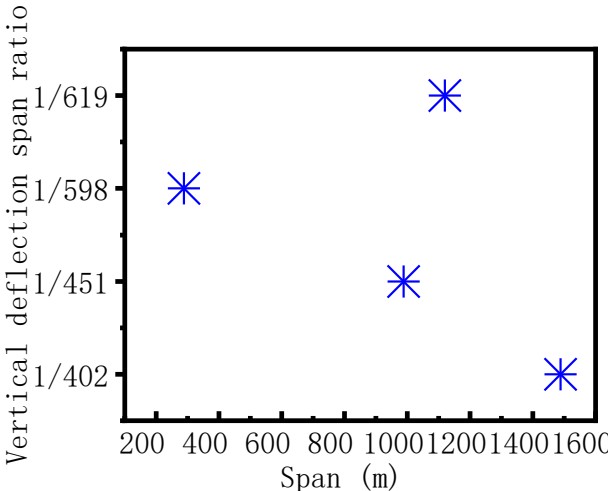

**Figure 18.** Investigation on engineering experience of vertical stiffness index.

Combining the requirements of driving safety and comfort and the existing engineering practice experience and considering a certain safety reserve, the limit value of the

deflection–span ratio of the cable-stayed-suspension cooperative system bridge is suggested to be 1/300. The designed vertical deflection–span ratio of the bridge should be 1/330 to meet the requirements of driving safety and comfort.

## 5. Conclusions

1. The evaluation of driving comfort and safety of bridges based on vehicle–bridge coupling vibration theory can be used as a reference method for determining bridge stiffness of long-span flexible bridges. Using random traffic simulation and a more accurate tire model is beneficial to improve the accuracy of road vehicle–bridge coupling vibration analysis.

2. The vertical stiffness difference between the suspension cable area and the stay cable area of the cable-stayed cooperative system bridge is obvious. The vertical acceleration response increases obviously when the vehicle travels to the transition section of the stay cable. The suspension cable section is more flexible than the stay cable section and more sensitive to the change of overall stiffness.

3. The bridge deflection ratio has a significant impact on vehicle ride comfort. The greater the deflection ratio, the worse the ride comfort index. From the point of view of ride comfort, the vertical deflection ratio of the single-tower space cable-stayed suspension cable cooperative system studied here is suggested to not be greater than 1/300.

4. On the basis of this study, the numerical analysis method proposed in this paper can be used to study the mechanical behavior of bridges under different loads and its influence on the driving performance in the future. A scaled model can be established for testing, providing a theoretical basis for the design of bridges with a cable-stayed-suspension cooperative system.

**Author Contributions:** Conceptualization, L.X. and Y.H.; methodology, X.W.; software, L.X.; validation, X.W.; formal analysis, Y.H.; investigation, Y.H.; resources, L.X.; data curation, Y.H.; writing—original draft preparation, L.X.; writing—review and editing, Y.H.; visualization, Y.H.; supervision, X.W.; project administration, L.X. All authors have read and agreed to the published version of the manuscript.

**Funding:** The support from the National Natural Science Foundation of China (Grant No. 52078424, 52178170), Natural Science Foundation of Sichuan Province (Grant No. 2022NSFSC0426), National Key R&D Program of China (Grant No. 2022YFB3706703) is gratefully acknowledged.

**Data Availability Statement:** No new data were created or analyzed in this study. Data sharing is not applicable to this article.

**Acknowledgments:** Thanks to all authors for their cooperation and innovation.

**Conflicts of Interest:** The authors declare no conflict of interest.

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
