# Peer review of "Study on Deflection-Span Ratio of Cable-Stayed Suspension Cooperative System with Single-Tower Space Cable"

_infrastructures, doi:10.3390/infrastructures8030062_

Round 1

Reviewer 1 Report

Generally, the quality of this paper is good. The results can provide potential references for the industry to determine the span length and some other structural parameters. Some editorial and technical issues are desired to be clarified before a publication is recommended.

1)It is recommended to include some statistical results in the abstract to show the industrial value of this work. The qualitative results look not very significant to highlight the value of this work.

2)It is desired to expand the literature review to point out the potential application of the present work to some other cable-stayed structures subjected to wind load or moving loads, like [1-4].

[1] Larsen, Allan, et al. "Dynamic wind effects on suspension and cable-stayed bridges." Journal of Sound and Vibration 334 (2015): 2-28.

[2] Duan, Fuchuan, et al. "Study on Aerodynamic Instability and Galloping Response of Rail Overhead Contact Line Based on Wind Tunnel Tests." IEEE Transactions on Vehicular Technology (2023). 10.1109/TVT.2023.3243024.

[3] Bryja, Danuta,. "Computational model of an inclined aerial ropeway and numerical method for analyzing nonlinear cable-car interaction." Computers & structures 89.21-22 (2011): 1895-1905.

[4] Song, Yang, et al. "Contact wire irregularity stochastics and effect on high-speed railway pantograph–catenary interactions." IEEE Transactions on Instrumentation and Measurement 69.10 (2020): 8196-8206.

3) Please give more details on how to obtain CS CL and CR. If these are from a previous paper, the reference should be given to facilitate the authors to follow.

4) Can any evidence be given to prove the validity of the present model? If one, please discuss the acceptance of the numerical accuracy.

5) Please clarify if only the vertical dynamics are modelled or if the whole spatial model is built.

6) A strange word ‘5. Conclusion’ appear following the third bullet of the conclusions.

Author Response

Dear Reviewer,

Thank you for taking the time to review our manuscript and for providing us with such thoughtful and insightful feedback. We appreciate the opportunity to revise our work in response to your comments.

After carefully considering your suggestions, we have made significant changes to our manuscript. We have addressed each of your concerns in detail and have provided additional clarification where needed. We believe that these revisions have strengthened the overall quality of our manuscript.

Please find attached a detailed response to your comments, which outlines the changes we have made and the rationale behind them. We hope that you will find our revisions to be satisfactory.

Once again, we would like to express our sincere gratitude for your invaluable feedback. We believe that your comments have helped us to improve the clarity, rigor, and relevance of our research, and we are grateful for your contributions.

Sincerely,

Lin Xiao, Yaxi Huang, Xing Wei

Comments and Suggestions for Authors

Generally, the quality of this paper is good. The results can provide potential references for the industry to determine the span length and some other structural parameters. Some editorial and technical issues are desired to be clarified before a publication is recommended.

Comment 1: It is recommended to include some statistical results in the abstract to show the industrial value of this work. The qualitative results look not very significant to highlight the value of this work.

Reply 1: Change the 16th line of the summary to 'Existing engineering experience shows that the vertical deflection-to-span ratio of cable-stayed bridges under live load is distributed between 1/400 and 1/1600, and the vertical deflection span ratio under the action of lane load is recommended based on numerical analysis.'

Comment 2: It is desired to expand the literature review to point out the potential application of the present work to some other cable-stayed structures subjected to wind load or moving loads, like [1-4].

[1] Larsen, Allan, et al. "Dynamic wind effects on suspension and cable-stayed bridges." Journal of Sound and Vibration 334 (2015): 2-28.

[2] Duan, Fuchuan, et al. "Study on Aerodynamic Instability and Galloping Response of Rail Overhead Contact Line Based on Wind Tunnel Tests." IEEE Transactions on Vehicular Technology (2023). 10.1109/TVT.2023.3243024.

[3] Bryja, Danuta,. "Computational model of an inclined aerial ropeway and numerical method for analyzing nonlinear cable-car interaction." Computers & structures 89.21-22 (2011): 1895-1905.

[4] Song, Yang, et al. "Contact wire irregularity stochastics and effect on high-speed railway pantograph–catenary interactions." IEEE Transactions on Instrumentation and Measurement 69.10 (2020): 8196-8206.

Reply 2: Thank you for your correction and suggestion. We have read the literature you provided and added the literature review to the 36th line of the first page. 'Both cable-stayed bridges and suspension bridges are classic cable-supported bridge systems, and the cable structure is prone to vibration. Bryja found that the cable car would exhibit unstable behavior under wind loads, such as violent swinging or sliding, when studying the interaction between aerial cable cars and cables [2]. The irregular shape of the contact line in the railway electrification system can lead to oscillation and resonance between the pantograph and the contact line, affecting the stable contact performance of the pantograph with the contact line [3]. Moreover, when the wind attack angle is at a specific value, the damping of the catenary becomes negative, and the running vibration reaches the maximum amplitude [4]. Both of the above structures are simple cable-supported structures, while bridge structures composed of multiple cables, such as cable-stayed and suspension bridges, are prone to vibration behavior under dynamic loads [5].'

Comment 3 : Please give more details on how to obtain CS CL and CR. If these are from a previous paper, the reference should be given to facilitate the authors to follow.

Reply 3: Thank you for your question. The values of CS, CL, and CR are based on the technical information report of the project provided by the design company, which is not detailed in this paper due to confidentiality considerations.

Comment 4 : Can any evidence be given to prove the validity of the present model? If one, please discuss the acceptance of the numerical accuracy.

Reply 4: Thank you for your question. This paper mainly relies on numerical analysis because the engineering project it is based on is still in the design stage and lacks experimental verification. However, the static and dynamic analysis results in Table 4 and Table 5 were respectively established in Midas and ANSYS by two researchers. The self-vibration characteristics of the models established in the two software have been strictly compared, and the correctness of the numerical analysis results is guaranteed.

Comment 5 : Please clarify if only the vertical dynamics are modelled or if the whole spatial model is built.

Reply 5: Thank you for your question. The entire spatial model was established in this paper. We singled out the vertical dynamics because the vertical force calculation method of the particle-damping-spring model in the past is significantly different from that of the vertical dynamics. To clarify this, we have changed the 136th line of the 4th page from 'vehicle model' to 'vehicle space dynamics model.'

Comment 6 : A strange word ‘5. Conclusion’ appear following the third bullet of the conclusions.

Reply 6: Thank you for your correction. This part is a spelling error. It has been corrected.

Reviewer 2 Report

This article presents interesting results of a research about a very important topic, which has not been sufficiently studied in the worldwide literature. Experimental validation in laboratory and real-life bridges using many different scenarios would be necessary to be shown in a second article, so that the promising results obtained so far could be applied in real-life cases in the future. The article is well written and the approach is innovative. Therefore, I recommend this article to be published after a minor revision:

* In some parts of the Abstract, it is not clear if the results are based only from a numerical model of a bridge or if some results are obtained from data taken of a real-life bridge; please clarify.

* Figure 1 must be improved, some numbers are incomplete/cut (e.g. “638 x 2”), whereas in the legend “cable-stayed 19 x 16” a space between “stayed” and “19” is required. Moreover, letters/numbers font and size should be changed for others more adequate.

* In caption of Figure 1, the name/location of the real-life bridge used for the numerical model should be included, as well as a picture in other figure.

* Introduction is very short, more references should be provided (only 3 references were included).

* Some references should be included in page 2 in line 64 and line 70, after “coupled vibration analysis theory of wind-vehicle-bridge” and “evaluation of track irregularity of high-speed railway”, respectively.

* An image of the ANSYS FEM model must be included in Section 2.1 (page 3 after line 97).

* In Figure 2 there are some words underlined in red as the software Microsoft Word does for misspellings, please correct that.

* Be sure that all the variables mentioned in Equations and Figures are described (explanation of the meaning) and the format is uniform (do not use lower case for a variable and then upper case for the same variable, italics/normal, different letter fonts/sizes, etc.), since there are several inconsistencies.

* There is a typo in page 4 line 141: “Figure.4” instead of “Figure 4”.

* In page 5, there is a mistake in line 168 and line 170: “??? is the force of the bridge on the vehicle” and “??? is the force of the bridge on the vehicle”. Different variables with the same meaning is wrong.

* In figure 5, “(a)” and “(b)” must be used for the two different diagrams and both of them should be briefly explained in the corresponding caption of the Figure. 

* In page 7 line 212, a reference for the RMSV should be provided.

* In page 7 line 221, the units for acceleration must be written correctly.

* In Figure 9, which is the parameter/units for the vertical axis?; and in Figure 10, which are the units for the horizontal axis (position)?

* In legend of Figure 10, seven “V” are shown (V1-V7), whereas in Table 2 there are only six “V” (V1-V6).

* The length units for the layout shown in Figure 11 must be indicated.

* In page 13 line 339, there is a mistake about the referenced Figure.

* In page 13 line 354, there is a mistake about the referenced Figure.

* In Figure 15, the parameter/units for the vertical axes must be indicated.

* In Figure 16, the vertical axis corresponds with “aw” instead of “comfort indicators”. The term “comfort indicators” can be used in the corresponding caption to mention that the parameter “aw” is a comfort indicator. Moreover, the meaning of the blue horizontal line must be mentioned.

* In page 17 line 439-440, there is a typo: “5. Conclusions”.

* In Section 5 (Conclusions) it would be convenient in the authors can explain the future works about this research in order to apply this methodology in real-life cases.

Author Response

Dear Reviewer,

Thank you for taking the time to review our manuscript and for providing us with such thoughtful and insightful feedback. We appreciate the opportunity to revise our work in response to your comments.

After carefully considering your suggestions, we have made significant changes to our manuscript. We have addressed each of your concerns in detail and have provided additional clarification where needed. We believe that these revisions have strengthened the overall quality of our manuscript.

Please find attached a detailed response to your comments, which outlines the changes we have made and the rationale behind them. We hope that you will find our revisions to be satisfactory.

Once again, we would like to express our sincere gratitude for your invaluable feedback. We believe that your comments have helped us to improve the clarity, rigor, and relevance of our research, and we are grateful for your contributions.

Sincerely,

Lin Xiao, Yaxi Huang, Xing Wei

Comments and Suggestions for Authors

This article presents interesting results of a research about a very important topic, which has not been sufficiently studied in the worldwide literature. Experimental validation in laboratory and real-life bridges using many different scenarios would be necessary to be shown in a second article, so that the promising results obtained so far could be applied in real-life cases in the future. The article is well written and the approach is innovative. Therefore, I recommend this article to be published after a minor revision:

Comment1: In some parts of the Abstract, it is not clear if the results are based only from a numerical model of a bridge or if some results are obtained from data taken of a real-life bridge; please clarify.

Reply 1: Thank you for your correction, 'The vertical deflection span ratio under the action of lane load is recommended based on numerical analysis' is modified to 'Existing engineering experience shows that the vertical deflection-to-span ratio of cable-stayed bridge under live load is distributed between 1/400 and 1/1600,  and the vertical deflection span ratio under the action of lane load is recommended based on numerical analysis'.

Comment 2: Figure 1 must be improved, some numbers are incomplete/cut (e.g. “638 x 2”), whereas in the legend “cable-stayed 19 x 16” a space between “stayed” and “19” is required. Moreover, letters/numbers font and size should be changed for others more adequate.

Reply 2: Thank you for your suggestion, which has been modified as required.

Comment 3: In caption of Figure 1, the name/location of the real-life bridge used for the numerical model should be included, as well as a picture in other figure.

Reply 3: In line 32 of The first page, 'The design scheme of a bridge across the Xunjiang River adopts a (638+638) m space cable single tower cable-stayed-suspension collaborative system scheme, the tower height is 231 m, and the spacing is 16 m,  with a total of 19 (Figure 1). 'describes the location of the bridge. The true full name of the bridge is not given due to confidentiality considerations.

Comment 4 : Introduction is very short, more references should be provided (only 3 references were included).

Reply 4: Thank you for your correction and suggestions. Due to the limited experience of this bridge type and the limited existing literature, the literature review in this paper is brief, and there are more references for numerical methods in the method introduction in the following paper. However, as suggested by you, the review has added 4 related research literatures. If more literatures are added to the review, the research focus of the paper will not be obvious.

Comment 5: Some references should be included in page 2 in line 64 and line 70, after “coupled vibration analysis theory of wind-vehicle-bridge” and “evaluation of track irregularity of high-speed railway”, respectively.

Reply 5: Thank you for your correction. This is an editing error. The previous reference was deleted by mistake. It has been added as "8. Deng Lu, He Wei, Yu Yang, & Wang Wei. (2018). Research Progress on Theory and Application of Coupled Vibration of Highway Bridges and Vehicles.  China Journal of Highway and Transport, (07), 38-54. "Harmony" 12. Chen, H., Zhang, W., & Li, J. (2020). Coupled dynamic response analysis of high-speed railway vehicle and track system under random track  irregularities. Journal of Vibration and Control, 26(19-20), 1855-1869. doi: 10.1177/1077546320919061 '

Comment 6: An image of the ANSYS FEM model must be included in Section 2.1 (page 3 after line 97).

Reply 6: Thanks for your suggestion, the image of ANSYS FEM model has been added as required.

Comment 7:In Figure 2 there are some words underlined in red as the software Microsoft Word does for misspellings, please correct that.

Reply 7: Thank you for your suggestion and correction. The picture has been modified as required. In Figure 3, The blue part represents the degree-of-freedom coupling, In Figure 3, the blue part represents the degree-of-freedom coupling, the red part repre-sents the force element, and the different numbers represent the different force elements,  number 5 in the figure represents the spring-damping force element, and number 253 represents the TMeasy force element,  and the green part is the constraint equation '.

Comment 8: Be sure that all the variables mentioned in Equations and Figures are described (explanation of the meaning) and the format is uniform (do not use lower case for a variable and then upper case for the same variable, italics/normal, different letter fonts/sizes, etc.), since there are several inconsistencies.

Reply 8: Thank you for your correction, we have rechecked the variable names in this article to make sure they are consistent. For example, line 166 on page 5 has the same F_(b,d)^v as the upper index in the picture; The subscripts CS(ψ), CL(ψ), and CR(ψ) have been written uniformly in line 177 on page 5.

Comment 9: There is a typo in page 4 line 141: “Figure.4” instead of “Figure 4”.

Reply 9: Thank you for your correction. It has been modified as required.

Comment 10: In page 5, there is a mistake in line 168 and line 170: “??? is the force of the bridge on the vehicle” and “??? is the force of the bridge on the vehicle”. Different variables with the same meaning is wrong.

Reply 10: Thank you for your correction, we have double-checked the variable names in this article to make sure they are consistent. Fb,d vchanged in line with the picture on page 166, and the subscript of 'CS(ψ), CL(ψ), and CR(ψ)' has been written in line 177.

Comment 11: In figure 5, “(a)” and “(b)” must be used for the two different diagrams and both of them should be briefly explained in the corresponding caption of the Figure.

Reply 11: Thank you for your correction. The two figures have been named as Spatial relation diagram of P point in FIG. 6(a) and Tire coordinate system in FIG. 6(b), and corresponding descriptions have been added in the article: The geodetic coordinate system is taken as the reference system (Figure 6(b)); The geodetic coordinate system is taken as the reference system (Figure 6(b)); Line X, Figure 6(a) shows a schematic diagram of the diagram this calculation method.

Comment 12: In page 7 line 212, a reference for the RMSV should be provided.

Reply 12: Thank you for your advice, References have been added '21. Lee, S., Lee, S., Lee, J., & Choi, S. (2018). A study on the evaluation of vehicle ride comfort using RMSV method. Interna-tional Journal of Automotive  Technology, 19(1), 69-76. '

Comment 13: In page 7 line 221, the units for acceleration must be written correctly.

Reply 13: Thank you for your correction. The modification has been completed as required.

Comment 14: In Figure 9, which is the parameter/units for the vertical axis?; and in Figure 10, which are the units for the horizontal axis (position)?

Reply 14: Thank you for your suggestion. The vertical axis of the original figure 9 (now Figure 10) represents the corresponding number of vehicles with different axle loads, while the unit of the horizontal axis of the original figure 10 (now figure 11) is m. To modify in the image.

Comment 15: In legend of Figure 10, seven “V” are shown (V1-V7), whereas in Table 2 there are only six “V” (V1-V6).

Reply 15: Thank you for your advice. This is a typo. I missed a line because of carelessness in those articles.

Symbol

Category

Ratio

Conversion coefficient

V1

Car

40.6%

1

V2

Bus

5.8%

2.0

V3

Buggy

13.8%

2.0

V4

Medium truck

10.9%

3.0

V5

Bid truck

10.1%

4.0

V6

Trailers

10.2%

4.0

V7

Container truck

8.6%

4.0

Comment 16: The length units for the layout shown in Figure 11 must be indicated.

Reply 16: Thanks for your suggestion, the layout unit (m) is now indicated in the drawing.

Comment 17: In page 13 line 339, there is a mistake about the referenced Figure

Reply 17: Thank you for your correction. Now the quotation of the picture at line 339 on page 13 has been corrected and it is on line 363 on page 13 of the body.

Comment 18:  In page 13 line 354, there is a mistake about the referenced Figure.

Reply 18: Thank you for your correction. The picture quote on page 13, line 354, has been corrected, and now it is on page 13, line 378.

Comment 19: In Figure 15, the parameter/units for the vertical axes must be indicated.

Reply 19: Thank you for your correction. Now the vertical axis unit of the picture has been explained in the picture. Since there are two curves in the picture, respectively representing acceleration and height, the units are not consistent, so the units are not explained on the vertical axis, but in the legend.

(a) Deflection-span ratio 1/330

(b) Deflection-span ratio 1/3030

(c) Deflection-span ratio 1/280

(d) Deflection-span ratio 1/250

(e) Deflection-span ratio 1/200

Comment 20: In Figure 16, the vertical axis corresponds with “aw” instead of “comfort indicators”. The term “comfort indicators” can be used in the corresponding caption to mention that the parameter “aw” is a comfort indicator. Moreover, the meaning of the blue horizontal line must be mentioned.

Reply 20: Thank you for your suggestion. We have changed the vertical axis of the picture to aw and indicated that the blue line is the limit value.

Comment 21: In page 17 line 439-440, there is a typo: “5. Conclusions’.

Reply 21: Thank you for your correction. This error has been corrected.

Comment 22: In Section 5 (Conclusions) it would be convenient in the authors can explain the future works about this research in order to apply this methodology in real-life cases.

Reply 22: Thanks for your suggestion, we have added in line 464 on page 17 of the conclusion. ' On the basis of the study in this paper, the numerical analysis method proposed in this paper can be used to study the mechanical behavior of Bridges under different loads and its influence on the driving performance in the future, and a scaled model can be established for testing, providing a theoretical basis for the design of bridge with cable-stayed suspension cooperative system.’

Round 2

Reviewer 1 Report

All my comments have been well addressed.